# Illegal Solid-Waste Dumping in a Low-Income Neighbourhood in South Africa: Prevalence and Perceptions

**DOI:** 10.3390/ijerph20186750

**Published:** 2023-09-13

**Authors:** Nobomi Ngalo, Gladman Thondhlana

**Affiliations:** Department of Environmental Science, Rhodes University, P.O. Box 94, Makhanda 6140, South Africa

**Keywords:** illegal solid-waste dumping, impacts, practices, prevalence, social justice

## Abstract

Illegal solid-waste dumping (ISWD) is prevalent globally with adverse social and environmental impacts, particularly in poor communities. Understanding the extent, practices and perceptions of ISWD is needed to inform interventions. Using GIS techniques and household surveys, this study examined the prevalence of dumpsites, and perceptions of ISWD in a low-income neighbourhood of Komani, South Africa. A total of 120 dumpsites were encountered in green spaces, empty residential plots and street edges, suggesting illegal dumping of solid waste is widespread. More than half of the respondents (58%) disposed of household waste in undesignated sites or burnt it, attributing this to the non-collection of waste. Potential interventions suggested were largely technical, including regular collection of waste, and the provision of bins and plastic bags. The paper suggests approaches to addressing sustainable solid-waste management should consider the views of local communities, who are principal stakeholders in the solid-waste production and management mix.

## 1. Introduction

Solid-waste generation is increasing globally [1], with serious implications on society and the environment. Worldwide, it is estimated that up to 2 billion tonnes of municipal solid waste were generated in 2016, and projections indicate up to 3.4 billion tonnes of municipal solid waste could be generated by 2050 as urbanisation and the population continue to grow [2]. Particularly, in the emergent BRICS (Brazil, Russia, India, China and South Africa) economies, it is predicted that rapid economic growth will result in the growth of the middle class characterized by high spending and consumption patterns, while rapid urbanization will compound waste generation per capita [3]. It is projected that by 2030, middle-income households will constitute about two-thirds of the world’s population, with the highest contributor being developing countries [4]. In general, there is a positive relation between economic growth and waste generation per capita. Thus, with projected economic growth in developing and BRICS nations, a rapid increase in waste generation is unavoidable. This means that municipal solid-waste management remains a key environmental issue for consideration in cities because poorly managed waste will have adverse direct and indirect impacts on human wellbeing, including the transmission of diseases, clogging of drains which results in floods, and pollution of both land and water bodies.

Compounding sustainable waste management is illegal solid-waste dumping, a sustainability challenge [3,4], particularly in developing country cities and towns [4]. Illegal solid-waste dumping has serious negative social, economic and environmental impacts on residents, especially in poor neighbourhoods [5,6,7]. The negative social and environmental impacts of illegal solid-waste dumping (ISWD) are diverse, including water pollution, poor air quality, unsightly environments, injuries to people, livestock deaths and poor health. For example, evidence shows that water bodies near dumpsites in Kolkata, India, were contaminated with cadmium and nickel, far exceeding the World Health Organisation limits for drinking water [5]. Triassi et al. [8] note the adverse impacts on air quality due to widespread ISWD practices in the Campania region of Southern Italy. People’s poor solid-waste management practices such as the open burning of waste can cause localised air pollution, while fire pits are a risk to children who often play in and around dump sites [5,9,10]. Further, ISWD in urban green spaces creates unclean and visual pollution in urban spaces [11]. Illegal solid-waste dumping on green spaces can impact livestock negatively through the ingestion of plastic and other poisonous substances which can lead to livestock disease or deaths [12]. A common feature in urban spaces is scattered waste which can lead to the blocking of waterways, and, in turn, breeding grounds for mosquitoes and rodents that carry diseases [13]. Economic impacts include low land and property values and low potential for ecotourism. Given the adverse social, economic and environmental impacts of ISWD to the well-being of residents, crafting pathways for addressing ISWD should be a priority for city planners, managers and public health authorities.

In many countries, legislative frameworks have been instituted but are insufficient to address ISWD due to other constraining factors such as a lack of financial resources, poor local infrastructure for waste-collection services, and inadequate planning and allocation of responsibilities [3]. In response, there is an increasing realisation providing waste-management services for all and making places liveable in the context of increasing municipal solid-waste generation which requires multi-stakeholder processes for engaging formal and informal actors in addressing ISWD [3,4], i.e., a recognition of the complexity of this environmental challenge. According to Kubanza and Simatele [9], social justice in all its manifestations can only be sought and ensured by utilising a rights-based approach that integrates the voices of poor people in the development of solid-waste management policies and strategies. Therefore, this means that the municipal institutions in charge of providing solid-waste management services have a social responsibility to ensure that all residents receive adequate services because it is their responsibility to protect them from adverse impacts associated with solid waste [4,9].

Central to solid-waste management is an understanding of the prevalence and resident’s perceptions of ISWD. An assessment of the prevalence and perceptions of ISWD can allow a better understanding of the extent of ISWD, as well as the overall effect on the environment and on human well-being. For example, a review of waste management in developing countries points to several reasons behind illegal waste dumping including poor infrastructure for the collection, transportation and disposal of solid waste, poor planning, limited financial results, limited skills and know-how, and negative public attitudes towards sustainable waste management [3]. In South Africa, empirical evidence suggests that lack of support infrastructure such as containers used for shared waste collection and 150 L black bins for waste disposal at the household level, inadequate waste collection and backyard dwellers are some of the reasons behind illegal dumping of waste [6]. Similar findings have been reported in Indian cities by Srivastava et al. [14]. Further, empirical standing on the distribution of illegal solid-waste dumps can ensure comparison between neighbourhoods, which can clearly illustrate the magnitude of inequality in service provisions among different demographic groups. An advanced understanding can be used to raise awareness about the problem of ISWD and inform strategies and management plans for the equitable sustainable management of solid waste in cities.

In South Africa, ISWD has been reported as a growing environmental problem [6,7,9,12,13]. South Africa produces roughly 122 million tonnes of solid waste per year, with households contributing at least 10% (about 12.7 million tonnes) of this waste [15]. The growth in solid-waste production is attributed to urbanisation, population growth, economic growth, and general human behaviour, consistent with emergent BRICS economies. With one of the highest urbanisations globally, it is feasible to expect that solid-waste production in South Africa is going to take an upward trend, which will exacerbate waste-management challenges in the country. It is estimated that about 31% (2015 levels) of South African households do not receive or refuse collection services [16]. Often, the most common option for waste disposal in unserviced communities is illegal solid-waste dumping (ISWD) [16]. In South Africa, ISWD in residential areas is a well-documented social and environmental problem but its impacts are mostly visible in low-income neighbourhoods [6,13,17,18]. South Africa has a marked history of segregated development emanating from Apartheid-era spatial development policies, with marked inequalities in service provision along economic and racial lines [9,13,18,19]. These inequalities are evident in many spheres of people’s lives, including inequalities in access to urban green infrastructure, health, residential space and other amenities [20,21]. Therefore, understanding the prevalence of ISWD is not only beneficial for service-provision planning but is also an environmental justice imperative. One of the first steps needed to address the inequalities is to advance an understanding of the prevalence, perceptions and practices of illegal solid-waste dumping in low-income neighbourhoods, as a basis for developing socially just and equitable waste-management programmes in cities [9].

From an environmental justice perspective, city officials have a mandate to render waste-collection services to all citizens for a habitable environment, but service delivery remains a huge social and political issue, particularly in poor neighbourhoods in South Africa [6,21]. However, though it is known that ISWD is widespread in poor South African neighbourhoods, this is rarely supported by empirical analyses of the prevalence and perceptions of ISWD. Empirical evidence is needed to design interventions. There are promising signs that municipal solid-waste management as a social and environmental justice issue is gaining traction in South Africa as evidenced by growing research on the subject, but a lot of this work is focused on perceptions of ISWD in major cities [6,13,19,21] with a few notable exceptions [22].

Examining the extent of ISWD can provide empirical evidence on the magnitude of the problem which can be used to leverage responsiveness from service providers and waste-management practitioners, while an understanding of perceptions can allow for the voices of the marginalised groups of people to be reflected in solid-waste management interventions [18]. With calls for polycentric governance in waste management [1,3], understanding the prevalence of ISWD can advance our understanding of the state of affairs among poor communities, with a view to considering poor households as principal stakeholders in efforts for municipal solid-waste management. South Africa presents a globally relevant case to study because, as part of the emergent BRICS economies, it is grappling with issues that can present complexities to sustainable solid-waste management due to rapid urbanisation and economic growth, and associated growth of the middle-income demographic group, and inequalities in solid-waste-collection services which can jeopardise the goal of achieving inclusive, clean, safe and sustainable cities. Given this background, the main objective of this study was to examine the prevalence and perceptions of illegal solid-waste dumping among low-income residents of Komani—a medium-sized town in South Africa. The key questions of the study included: (1) what is and what are peoples’ perceptions of the prevalence of ISWD (2); what are the reported solid-waste-disposal practices; and (3) what are people’s perceptions of the impacts of ISWD and potential interventions.

## 2. Materials and Methods

### 2.1. Study Area

The study was conducted in Komani (previously Queenstown), a medium-sized town located in the Eastern Cape province of South Africa (32°16′10″ S, 27°15′24″ E). The town was founded in 1853 by colonial settlers. Komani is one of the largest towns under the Enoch Mgijima Local Municipality with a population of nearly 70,000 people [23]. It is a social and economic hub for industrial activity, commerce, and social-development services [24]. Consistent with trends in the Eastern Cape province, the poorest province in South Africa [25], Komani, is characterised by high levels of poverty with up to 61% of the population living below the poverty line [24], low education levels with a literacy level of less than 50% and high unemployment rates [26]. Mirroring patterns in other South African towns, the geographical setting of Komani has imprints of Apartheid-era planning. In keeping with Apartheid policy of separate development by race, most black South Africans were moved to the eastern side of the town. Thus, the western side of the city, formerly reserved for whites, is characterised by large residential plots, big gardens, and large green spaces, including parks and sports fields. It is generally home to well-off households. By contrast, the eastern and southern sides are inhabited by poor households with a low level of education. Residential plots are relatively small, houses are small, and amenities are either few or dysfunctional. In general, municipal service provision, such as waste removal, is poor or non-existent in the low-income neighbourhoods, mirroring trends elsewhere in the country. Records suggest about 36% of the population does not receive municipal waste-collection services [26], largely owing to dysfunctional municipalities and poor service delivery, with this problem likely to be more widespread in poor neighbourhoods than in well-off ones. The town of Komani has faced poor service delivery dating back to 2018 when the local municipality’s maladministration practices emerged. Following a High Court order to pay civil engineers who had rendered services to the municipality, it is claimed by media reports [27], that the municipality auctioned several assets including garbage-collection trucks to raise the funds. In January 2023, the Komani residents held several protest actions against poor service delivery including power outages due to dilapidated electrical infrastructure, potholes, and non-collection of refuse in the municipality—which forced the Minister of Cooperative Affairs and Traditional Governance to travel to the town to resolve issues behind the protests [28].

In South Africa, municipal solid-waste management is regulated by the National Waste Management Strategy 2020 in terms of Section 6 of the National Environmental Management: Waste Act, 2008 (Act No. 59 of 2008) premised on three pillars, namely, waste minimisation, effective and sustainable waste services and enforcement and awareness. These pillars are meant to pave “a future South Africa with zero waste in landfills, cleaner communities, well-managed and financially stable waste services and a culture of zero tolerance of pollution, litter and illegal dumping”, p8 [29]. The National Domestic Collection Standards No. 32,687 is aimed at addressing historical imbalances in waste-collection-service provision by making sure waste-collection services are acceptable, affordable and sustainable as a basis for improving the quality of life and providing a clean and liveable space for South Africans.

### 2.2. Data Collection

Data collection took place in July 2022 in four southern neighbourhoods of Komani, namely Mlungisi, Newrest, Victoria Park, and Aloevale. The study involved two steps. Step one identified and mapped solid-waste dumps in the study sites, and step two collected information on residents’ perceptions and practices regarding illegal solid-waste dumping.

Step one: Street and off-street surveys targeting all streets in the study sites were used to identify illegal solid-waste dumps. For each dump encountered, the following information was collected: the coordinates of the dump, the location of the dump (open area, empty residential plots or street edge), the size of the dump, the solid-waste composition of the dump and any notable activities on the dump (e.g., evidence of fire, children playing on dump and foraging domestic animals). The coordinates of each solid-waste dump were recorded using a global positioning system (GPS) device. Information on the location of each illegal solid-waste dump, dump size, solid-waste composition, and key activities on the dump and visible notices prohibiting illegal solid-waste dumping was captured through direct observations and photos. The size of dumps was visually estimated using a subjective scale (large, medium, or small). Large dumps extended for more than 30 m in length, medium-sized dumps were between 5 metres and 30 metres, and small dumps were less than 5 m. The type of solid-waste materials making up the dump was determined using a widely available categorisation (household waste, construction waste and garden waste) [22].

Step two: The second step was a qualitative survey conducted in the four neighbourhoods using a semi-structured questionnaire (Appendix A). A randomised approach using GIS was employed to select 200 households for the surveys. The randomisation process involved the identification of homesteads in the selected sites using google maps. The homesteads were allocated numbers, and a simple selection of target households was conducted through the use of the RANDBETWEEN (1,2) function in Microsoft Excel. The randomly selected households were identified for interviews in the field. If the randomly selected household was unavailable, the next available household was approached for the surveys. However, due to absent members in a substantial number of randomly selected households (due to work and family or social commitments), a systematic approach was later employed, where every fifth house from the last interviewed household was targeted for the interviews. Household heads were targeted for the interviews, and if unavailable, the eldest member of the household. The interviews were conducted in the local IsiXhosa language and took between 30 and 45 min each. Ethical clearance for the study was granted by the Rhodes University Human Research Ethics Committee (Approval Number 2022–5620-6719).

The questionnaire was divided into two sections with open and closed questions. The first section collected social demographic data of the respondents and their households including gender, age, education level, occupation, income and household size. The second collected respondents’ perceptions of the prevalence of illegal solid-waste dumping in the study areas, its causes and impacts, solid-waste-disposal practices, the frequency of the disposal practices and whom they thought was responsible for addressing illegal solid-waste dumping.

### 2.3. Data Analysis

GIS spatial maps and descriptive statistics were used to show the abundance, distribution and intensity of illegal solid-waste dumps that were encountered. Clusters with four or more dumps in each cluster and unclustered dumps with a single-point feature were presented to show the distribution and intensity of illegal solid-waste dumps. Visual maps were used to indicate areas with a high and low density of dumps at a neighbourhood zoom-out visible scale range of 1:20,000. The density of dumps by neighbourhood was illustrated using a heat map, with dark colours representing a high density of dumps and light colours representing low density. Descriptive statistics were used to summarise the demographic profile of the respondents, frequency of responses regarding reported solid-waste disposal practices, perceived factors contributing to and impacts of illegal solid-waste dumping and potential interventions. A chi-squared test was performed to check for association between gender and waste disposal practices reported by the respondents. Thematic analysis was used to identify the factors that contribute to illegal solid-waste dumping, the perceived impacts of illegal solid-waste dumping, and potential interventions. Direct quotes from the respondents were used to support quantitative data and provide qualitative lenses for drawing and expressing meaning from the findings.

The limitations of this study relate to potential sample-selection bias [30] due to the non-probability sampling strategy employed (following failure to obtain respondents in randomly selected households) and the inability to generalise the findings due to the small sample size [31]. Despite these limitations, the study offers potentially useful and transferrable insights on the prevalence of illegal solid-waste dumping which can be used to better understand and advocate for action to address the problem.

## 3. Results

### 3.1. Abundance, Distribution, Size and Composition of Illegal Solid-Waste Dumps

Figure 1 indicates the distribution and size of the identified solid-waste dumps. A total of 120 dumpsites were encountered. Out of all the dumps encountered, nearly half (45%) were in green spaces (mainly urban parks) while the remainder were in open spaces including street edges, around public infrastructure sites such as ESKOM transformer sites (10%), empty residential areas, and edges of public service infrastructure such as schools and clinics. A sizeable number of the dumps were classified as large (63%), followed by medium (21%) and small (16%). When considered by neighbourhood, out of all the dumps encountered, 54% were located in the low-income area of Mlungisi (54%) than the low- to middle-income areas (46%). Despite notices prohibiting illegal solid-waste dumping in certain areas, dumps were encountered in these prohibited areas, suggesting that the legislation was ignored or not enforced.

Figure 2 displays groups of illegal solid-waste dumps within a distance from one another using the cluster image displays (with four or more dumps in each cluster). Unclustered dumps sites are shown as a single-point feature. The results show 11 clusters made up of at least four dumps. Figure 3 indicates the intensity of illegal solid-waste dumps by neighbourhood. The results show a higher density of illegal solid-waste dumps in the low-income neighbourhood of Mlungisi than in the medium-density neighbourhoods.

The solid waste encountered in illegal dumps was diverse and included typical household waste (plastic, paper, glass, old clothes, and food), dead domestic animals such as dogs, pigs and cats, garden waste and construction waste. Almost all the dumps encountered consisted of household waste (97%), garden waste (91%) and construction waste (66%).

The results suggest that illegal solid dumping is widespread, with solid-waste dumps encountered in empty residential plots, recreational areas and street edges. This is corroborated by the fact that open dumping and burning were the main methods of household solid-waste disposal reported by the respondents. These findings are consistent with findings by Mngomezulu et al. [7] in Nelson Mandela Bay, South Africa and Southwest Nigeria. Solid-waste dumps are often found in these places because of poor service delivery (waste-collection services) by dysfunctional municipalities’ failure to provide adequate services and infrastructure for solid-waste removal. The findings are consistent with findings by Aluko et al. [32] in Nigeria, who found that the majority of residents (85.6%) attributed ISWD to the scarcity of garbage-removal trucks.

In both developing countries and developed countries, a combination of rapid population growth, high poverty rates, and urbanisation can result in high solid-waste production. Particularly in low-income neighbourhoods, characterised by high-density settlements and poor waste-service infrastructure and collection, environmentally unfriendly waste-disposal practices are common. Pineo and Rydin [33] show a positive correlation between socioeconomic factors and solid-waste management. For instance, where there is no infrastructure provision such as bins that could help steer people towards good behaviour, residents tend to resort to unsustainable disposal practices, including open dumping and burning of waste. In this study, dumps were encountered in open spaces, street edges, recreational areas, and green spaces, likely creating a breeding ground for insects, pests, and rodents—key vectors of infectious diseases [8], with negative impacts on local residents’ well-being.

Illegal solid-waste dumping is linked with health and environmental impacts including visual pollution of degradable and non-degradable materials (e.g., diapers, plastic material, left-over hazardous health waste such as flammable products like methylated spirits and paint thinners, and corrosive materials such as batteries and oven and toilet cleaners).

### 3.2. The Socio-Demographic Profile of the Respondents

Table 1 summarises the socio-demographic profile of the respondents. Out of 176 respondents interviewed, nearly two-thirds were female (62%). The average age of respondents was 54 ± 16 years. The mean household size for the sample was about five people.

Nearly two-thirds of the respondents had achieved a secondary-level education and about 15% had either no education or primary education. Only 20% of the respondents had tertiary-level education. More than half of the respondents were either unemployed (31%) or pensioned (30%), while just about a fifth were employed and the remaining proportion was self-employed. Out of all the income sources mentioned, a substantial proportion of the respondents depended on social welfare grants (38%) with wages, pensions and income from their own business also being key income sources. Just about half (51%) of the households received an average monthly income of less than ZAR2000 (USD 123) and a further 30% between ZAR2000 (USD 123) and ZAR6000 (USD 927), indicating high poverty levels. On average, respondents had lived in the area for about 43 ± 23 years.

### 3.3. Solid-Waste-Disposal Practices

When asked about their waste-disposal methods, over one-third (36%) of the respondents reported open disposal as a highly practiced disposal method (Table 2). Other disposal methods mentioned include burning (22%), backyard disposal (10%) and landfill. Only one household mentioned recycling as a waste-disposal method. No one mentioned the use of municipal refuse-collection services. There were no significant differences in waste-disposal methods between males and females (χ^2^ = 3.919; *p* = 0.561). In the study site, residents reported that refuse-removal trucks did not come on scheduled days and times; hence, the residents resorted to dumping their solid waste on open sites, arguing that they did not have enough space to keep the rubbish, were unable to stand the smell of foul rubbish and if they left the waste outside their yard livestock (donkeys, goats and pigs) will forage on the solid waste, which left an unsightly impression.

### 3.4. Factors Contributing to Illegal Solid-Waste Dumping

When asked what factors contributed to illegal solid-waste disposal, nearly half (48%) of the respondents reported poor waste-collection services (Table 3) citing a ‘dysfunctional municipality’. The respondents explained that poor service delivery was the main reason behind illegal solid-waste dumping, highlighting that refuse trucks did not collect refuse as per schedule due to breakdowns, few refuse trucks and limited human resources to do the job. Referring to persistent refuse-truck breakdowns one respondent said: “When trucks stop working, the waste does not stop growing”. Other respondents cited a general lack of care among residents (35%) attributing this to lack of service delivery and lack of resources to support waste-collection activities (Table 3). About 7% of the respondents said that lack of resource provision such as refuse bags, refuse bins and refuse drop-off areas contributed to ISWD in the area. Lack of awareness about the social and environmental impacts of ISWD, municipal strikes due to perceived corruption and nepotism, and poor service delivery in general were also cited as reasons behind ISWD, though this was mentioned by a small proportion of the respondents.

The reported factors behind illegal solid-waste dumping can be classified as administrative and behavioural. From an administrative perspective, the results show many respondents attributed ISWD to a dysfunctional municipality that failed to fulfil its service-delivery mandate. For example, the residents cited an absence and selective provision of support infrastructure such as bins, bin liners and skips. Institutional or administrative failure is an already standing issue in South Africa, particularly in towns and cities of the Eastern Cape province, which can be traced back to pre-1994 [6,19]. It is also common to find disproportionate service delivery between well-off and poor communities, with the former receiving more services from local municipalities than the latter, a pattern that is attributed to Apartheid-era spatial development policies, which promoted and entrenched inequalities in service provision based on race and economic status [15,19,20]. Patterns of unequal access to service provision have also been highlighted by Polasi [34] who cites inequalities in infrastructure and service delivery in low-to-middle income communities. Schenck et al. [21] also found that municipal failures to abide by the weekly schedule of solid-waste collection due to old and outdated refuse trucks (which are not fit for purpose) was behind illegal solid-waste dumping in Philippolis, South Africa. Institutional factors such as municipal failure can be attributed to incompetence and rampant corruption which is widely reported to interfere with adequate service provision in South Africa [6,13]. Although this study did not explore corruption as a factor, many residents perceived corruption as one of the main reasons behind municipal failures not only in the study area but also in other municipalities in the country. Many residents interviewed stated that the lack of adequate infrastructure was due to the misuse of waste-management funds, which could have been used to potentially improve existing infrastructures. This, in turn, resulted in feelings of powerlessness to control the illegal solid-waste-disposal problem on the part of the residents. Non-caring attitudes are socially related, particularly in fragmented communities due to structural poverty, the South African historical background, and political and economic landscapes [21], which arguably can prevent people from addressing solid-waste issues. However, it is plausible to suggest that the challenge of ISWD could be explained by other factors such as lack of financial resources, inadequate planning and ill-informed allocation of responsibilities as has been found elsewhere in South Africa [6] and beyond [3,35]. Particularly, in BRICS and developing countries, inadequate planning and insufficient budgets have been reported as key constraints to rendering acceptable waste-management services [35].

There is a correlation between context and behaviour, where situation-based factors can result in norm-violating behaviour such as illegal dumping of solid waste, as made evident in this study [36]. Situational factors explain the ‘I don’t care’ attitude reported by the respondents because if people feel they do not have the power and voice to change the situation, then they are unlikely to act in the interest of the environment [9,21,36]. Having control over the situation is positively related to pro-environmental behaviour [37].

Administrative failures and behavioural factors are linked to negative social behaviour beyond waste-disposal practices. People tend to blame other people for not caring and taking initiative for change. Hansmann and Steimer [36] identified a common trait of littering in Switzerland that is often attributed to external causes. Participants rarely blamed themselves but blamed others for the high prevalence of ISWD, exhibiting self-serving bias for socially disapproved behaviours such as illegal waste disposal [36]. In this study, most respondents did not highlight their responsibility for the high prevalence of ISWD as residents justified their action as entitled rate payers and blamed others. For instance, middle-income residents in Aloevale, South Africa blamed others (newcomers) from low-income neighbourhoods for ISWD in their neighbourhoods. In Paarl East and Mbekweni, South Africa, Schenck et al. [21] similarly found that the respondents blamed ‘migrants’ for the prevalence of illegal dumpsites and did not identify their own role in causing and addressing the problem.

### 3.5. Perceived Impacts of Illegal Solid-Waste Dumping

Table 4 shows the perceived impacts of ISWD by residents. Most respondents (70%) felt that ISWD reduced the economic value of their properties and neighbourhood, citing that they seldom considered selling their properties because no one found the area attractive for living or investment. About 61% of the respondents reported loss of visual amenity due to uncollected solid waste in different areas of their neighbourhood including green spaces, public service facilities such as schools and clinics, and street edges. Others mentioned poor air quality (25%) from burning solid waste and foul smells from decaying waste, which negatively impacted their daily routines and, in turn, compromised their well-being. For example, some respondents reported spending more time indoors than outside and always shutting windows and doors to avoid unpleasant smells in their homes. Evidence of fire (burning fires or ash) was observed in nearly three-quarters (75%) of all the encountered dumps. Health hazards to both animals (19%) and people (15%) were also reported, with respondents citing polluted water sources such as streams and blockage of drainage systems. The respondents felt that this, in turn, caused flooding and spreading of diseases, injuries to children from broken glass, infestation of disease vectors such as rats and flies and sickness, and death of livestock. Direct observations showed that children played on dumps, livestock, especially pigs and goats foraged at dumps, and dogs scavenged on dumps. Others felt that ISWD negatively impacted business in their neighbourhoods because no one wanted to invest in a dirty environment.

Illegal solid-waste dumping was perceived to have negative social and environmental impacts on residents. Reported social impacts stemmed from localised air pollution from fire pits, which negatively impacted air quality and subjected children who played at dump sites to danger and health risks. In Haiti, Medgyesi et al.’s [10] findings show children frequently play with and touch waste, and put their hands in their mouths, resulting in the ingestion of traces of human and animal faecal matter capable of causing gastrointestinal infections. Viljoen et al. [22] found children playing at dumpsites as a challenge, highlighting parents’ lack of responsibility in ensuring their children do not play on dumps. The reported practice of open dumping and burning of waste is a potential threat to the quality of the environment due to wind-blown litter located along fences, and smoke spreading into people’s yards. The perceived environmental impacts were not only the degradation of green spaces such as recreational parks and unclean environments but also a health hazard for people and death of animals. Highly cited health consequences by participants were consistent with the findings by Dzawanda and Moyo [11], who highlighted that burning of waste aggravated respiratory, cardiovascular, and skin irritation. In this study, several domestic animals including pigs, goats, dogs and cows were observed foraging at dumps, and dead animals were encountered in dumps. In a study of animal visitation patterns and foraging behaviour at dumpsites in Uttarakhand Himalaya, India, Katlam et al. [38] observed high feeding frequency of domestic and wild animals at waste dumps. The prevalence of domestic animals at dumps raises concerns about the health of livestock which can ingest hazardous waste such as plastic [38], as livestock rearing is important for residents for income generation and investment [3,12].

### 3.6. Potential Interventions Reported by the Respondents

Concerning potential interventions for addressing illegal solid-waste dumping, approximately 28% of the respondents believed the municipality should invest in efficient and effective solid-waste management strategies, including establishing refuse-disposal points and regular refuse collection. Other respondents mentioned provision of resources (14%) citing that the provision of bins and bin liners could promote environmentally friendly solid-waste disposal. Other interventions mentioned by the respondents include raising awareness about the benefits of recycling (13%), employing unemployed youths to clean up (11%), institution of penalties to offenders (7%) and fencing off open spaces (10%).

According to Patel [39], efforts addressing social and environmental justice in the context of solid-waste management must be viewed as inextricably linked because both concepts emphasize the importance of empirical understandings grounded in local contexts. Further, the focus of both social and environmental justice is to provide equal service delivery in solid-waste collection and disposal as a means for correcting previous injustices.

Investing in waste-collection infrastructure such as waste-removal trucks and bins will promote proper solid-waste disposal and regular collection. Adequate waste-collection facilities are required for efficient waste-management processing [5]. A pro-poor and equitable approach to waste management that prioritises poor neighbourhoods and increases the frequency and timely collection of solid waste due to a high density and production of waste is needed. Successful examples exist. For example, Schenck et al. [21] report a success story in Drakenstein Municipality, South Africa, where weekly day-to-day refuse collection and daily drop-off waste-collection services led to the municipality being known as the cleanest and greenest municipality in 2019.

Participants highlighted the need for efficient and effective municipal management such as keeping up a frequent and timely collection of waste schedules in all neighbourhoods because municipalities have an important role in managing and preventing ISWD. Participants reported that to mitigate ISWD, effective techniques such as raising awareness and educating people about recycling, and reusing and composting bio-degradable waste were potential interventions. With the high unemployment rates, participants found it fitting for the municipality to hire unemployed youths from their neighbourhood as this would not only create jobs but also build connections between people and nature, which is known to lead to pro-environmental attitudes [40,41].

Recycling has numerous social, economic, and environmental benefits, and its potential in South Africa must be developed and prioritised [42]. Findings by Joel and Fansen [43] show that while waste composition varies by location and income group, most waste including paper, cardboard, glass and plastic generated by inhabitants in developing countries is recyclable and can provide economic benefits, particularly to low-income groups. For example, it is estimated that up to 80% of the 125 million tonnes of solid waste generated per year in Africa is recyclable but only 4% is recycled. In South Africa, of the 59 million tonnes of solid waste produced, only 10% is recycled, suggesting there is scope for recycling as a potential pathway towards sustainable solid-waste management [44]. Recycling can create employment in low-income communities. Local municipalities could employ residents to keep the town clean by providing education on environmental clean-ups and recycling programmes, as well as starting awareness campaigns, which can engage the community about innovative ways for resolving waste-management issues [6,17]. Recycling is a potentially useful pathway considering that only few households actually recycled their household waste, consistent with findings by Plastics SA [44,45] that indicate a recycling rate of just 10% at the national level. According to Plastics SA [45], South African plastic recycling manufacturing provided nearly 58,750 people with informal income opportunities in 2019, meaning that further investments in recycling can reduce the amount of waste and provide employment opportunities, which can be a key livelihood contribution to marginalised communities struggling with high unemployment. Recycling strategies should integrate informal waste pickers as studies suggest their role in waste management systems is more important than appreciated, yet they are continuously neglected in the formulation of waste-management policies and strategies [46,47]. For example, the provision of market incentives has been linked to increased solid-waste recycling in Bangladesh [48], meaning the integration of the informal collectors coupled with support systems can encourage recycling. Overall, solutions lie in integrative strategies rather than relying on a single intervention to mitigating illegal solid-waste dumping as has been suggested elsewhere [49,50].

## 4. Conclusions

The research shows that illegal solid-waste dumping is widespread in low-income neighbourhoods. Poor service delivery by the local municipality seems to be central to the problem of ISWD. Given the history of a segregated approach to urban development, it is plausible to argue that poor service delivery by the local municipalities has disproportionate impacts on the urban poor, making addressing ISWD a social justice issue. For poor communities, ISWD dumping can have far-reaching impacts on well-being because it takes place closer to homes, in recreational facilities meant to serve these communities and on public infrastructure such as electricity substations. Since these communities often cannot afford these alternatives, the impacts of ISWD should be serious concern for urban authorities, planners and practitioners.

Addressing ISWD in towns requires a multi-faced approach, where local municipalities are capacitated to improve waste collection and management, and residents are sensitised about their own role in addressing the problem. The involvement of the informal sector in addressing ISWD in cities offers promise but optimum benefits can be achieved if the role of the informal sector is recognized in waste management policy and support systems are in place such as the relevant infrastructure, market incentives and protective equipment. Other options, such as recycling, offer opportunities but require huge capital investment to establish the infrastructure and residents’ willingness to separate waste at source (household level). Given the disproportionate impacts of poor service delivery on poor communities, employing a pro-poor approach to addressing ISWD might assist in addressing the problem of waste dumping and making poor neighbourhoods safe and healthy places to live. In designing potential interventions, it is important to consider that interventions solely designed by the government as a service provider often fails, but the co-designing of interventions by diverse stakeholders, such as government, the private sector, waste-management practitioners and householders, might yield optimum results, as proposed elsewhere [1]. These potential interventions will yield the desired outcomes only if present barriers such as poor waste infrastructure, poor planning and lack of skills are addressed.

## Figures and Tables

**Figure 1 ijerph-20-06750-f001:**
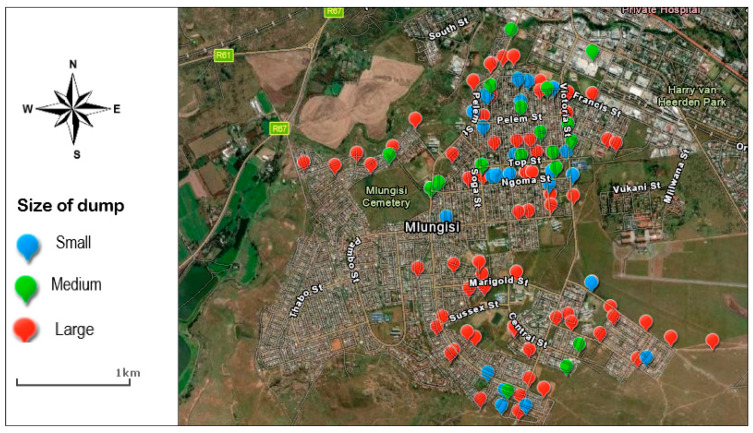
The distribution and size of illegal solid-waste dumps.

**Figure 2 ijerph-20-06750-f002:**
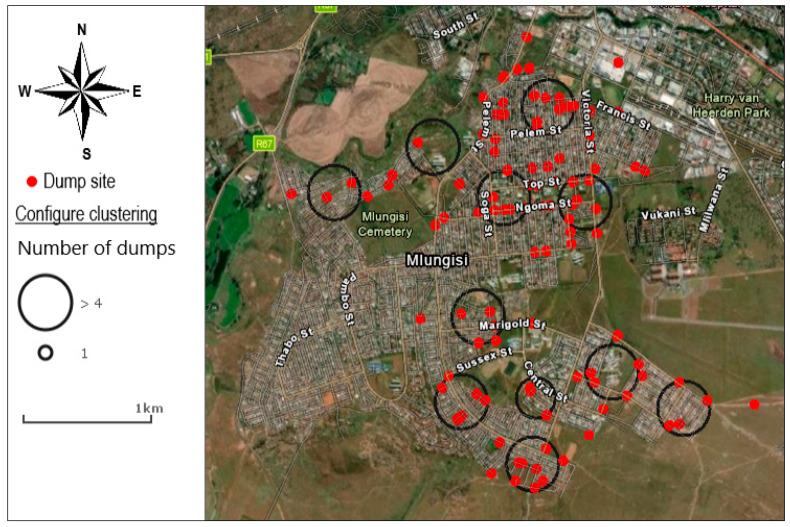
The density of illegal solid-waste dumps.

**Figure 3 ijerph-20-06750-f003:**
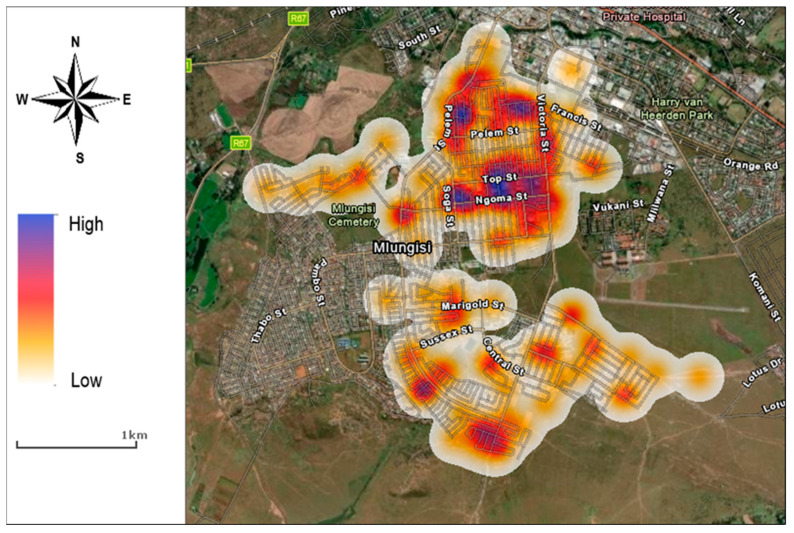
Variation of density of illegal solid-waste dumps by neighbourhood.

**Table 1 ijerph-20-06750-t001:** Socio-demographics of the respondents.

Household Socio-Economic Factors	Values
**Gender (%) of household head**	
Female	62
Male	38
Mean age (±SD) (in years) of household head	54 ± 16
Mean (±SD) household size	5 ± 3
**Education level (%)**	
No education	1
Primary school	14
Secondary school	65
Tertiary	20
**Average monthly income (ZAR) (%)**	
<2000	51
2001–6000	30
6001–15,000	9
15,001–30,000	7
>30,000	3
**Mean (±SD) length of stay**	43 ± 23

**Table 2 ijerph-20-06750-t002:** Reported waste-disposal methods.

Disposal Method	Proportion (%) of Respondents
Open disposal	36
Burning	22
Backyard disposal	10
Landfill	5
Recycling	1
Others	26

**Table 3 ijerph-20-06750-t003:** Factors contributing to illegal solid-waste dumping.

Reported Factor	Proportion (%) of Respondents
Poor waste collection services	48
Lack of care	35
Lack of resources	7
Lack of awareness of impacts	4
Municipal strikes	3
Others	4

**Table 4 ijerph-20-06750-t004:** Reported impacts of illegal solid-waste dumping.

Impacts	Proportion (%) of Respondents
Loss of property value	70
Visual pollution	69
Poor air quality	25
Livestock sickness and deaths	19
Health hazards to people	14
Business impacts	11

## Data Availability

The data that support our research findings are available from the corresponding author on request.

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
