# Peer review of "Illegal Solid-Waste Dumping in a Low-Income Neighbourhood in South Africa: Prevalence and Perceptions"

_ijerph, 2023, doi:10.3390/ijerph20186750_

Round 1

Reviewer 1 Report

It is an interesting subject and a methodology that could be replicated in other countries with similar characteristics to allow comparison. There are two weaknesses: 1) theoretical support of the analysis. We suggest that you continue to include some references that discuss or issue, such as in SILVA, Christian Luiz da; WEINS, N.W.; POTINKARA, M. . Formalizing the informal? A perspective on informal waste management in the BRICS through the lens of institutional economics. WASTE MANAGEMENT, v. 99, p. 79-89, 2019 (https://www.sciencedirect.com/science/article/abs/pii/S0956053X19305422?via%3Dihub).

2) The methodological question is limited by the number of interviewees. This should be indicated as a restriction or, if necessary, which sample level calculated. Furthermore, the methodology should present the research stages and techniques that could be used in case of wanting to replicate it in another region.

Finally, the conclusions should draw more pragmatic elements about what makes it difficult for this multi-faced approach to be applied. Despite being described in a linear way, there are few difficulties in making it operational. These challenges should be posted.

Reviewer 2 Report

This study analyzed the prevalence of dumpsites and perceptions of ISWD in a low-income community of Komani, South Africa, using geographic information system technology and household surveys. The identification of 120 dumpsites in green Spaces, vacant blocks and street edges in Komani low-income communities in South Africa shows that illegal dumping of solid waste is widespread, with more than half of respondents (58%) disposing of household waste in unspecified locations or burning it, blaming community failure to collect waste. Potential interventions suggested in the article are largely technical and include regular waste collection and the provision of bins and plastic bags. The topic of this article has certain practical significance, but there are some problems in the structure of the article, so it is suggested to publish after modification.

Revise opinions:

1.     The introduction is not recommended to describe in sections.

2.     Two parts 2.2 appear in the text, please renumbered.

3.     2.2 Data collection involves a variety of methods, which are listed and introduced separately in the Materials and Methods section.

4.     2.2 and 2.3 of the Materials and Methods section describe the idea and process of organizing the article. The description of the overall idea of the article should be briefly described in the last paragraph of the introduction.

5.     Parts 3.1 and 3.2 can be merged into one part, and it is suggested to make a table of garbage types.

6.     The third and fourth chapters of the article are suggested to be directly merged into one part, named results and discussion, and the results should be analyzed and discussed directly after the survey results are listed in this part, so as to make the article easier to read.

7.     How was the heat map of Figure 3 in the text generated?

The quality of English is basically passed.

Author Response

Please, see attached file.

Reviewer 3 Report

My main comments for this study will be related to methodology. First, I suggested studying the steps shown in the flow diagram. The second questionnaire was added in article or as a supplementary file. Figures related to the study area and mapping are not high quality and it had to draw again with good quality and some brand writing with acceptable font and size. Also kind of mapping method by GIS must be mentioned and added RMSE.

My main comments for this study will be related to methodology. First, I suggested studying the steps shown in the flow diagram. The second questionnaire was added in article or as a supplementary file. Figures related to the study area and mapping are not high quality and it had to draw again with good quality and some brand writing with acceptable font and size. Also kind of mapping method by GIS must be mentioned and added RMSE.

Author Response

Please, see attached file.

Reviewer 4 Report

Comments: Illegal solid waste dumping in a low-income neighborhood in South Africa: prevalence and perceptions

Introduction

1.- The bibliographic citations are very limited because only 5 are mentioned. In addition, they are used repeatedly in the different paragraphs. For example, reference 3 is cited four times in the introduction.

2.- Line 35 indicates that "…Illegal solid waste dumping has serious negative social and environmental impacts…" but the economic and legal impacts (if applicable) are not mentioned, so it is also necessary to clarify whether there are federal laws or premises that regulate waste management.

3.- Line 52 mentions "ISWD should be a priority for city planners and managers...", but the public health authorities would not be mentioned.

4.- In lines 60 to 62 it is mentioned "…Assessment of the prevalence and perceptions of ISWD can allow a better understanding of the extent ISWD, as well as the overall effect on the environment and human wellbeing..." but it is not indicates what has been researched on the subject by different authors.

5.- At the end of the introduction, the objective of the work must be indicated.

Materials and methods.

6.- This section omitted to present the methodology of the results presented in sections 3.2 to 3.7

7. The study area paragraphs are likely to be separate from the methodology. This information could be incorporated into the results section.

8.- On line 132 they indicate that 36% of the population does not receive municipal services, but they do not clarify the reason(s).

9.- In line 149 they refer to the research question, but they do not indicate it anywhere in the manuscript.

10.- How will the sample size indicated in line 155 be extended, where they indicate a total of 176 households surveyed?

3. Results

11.- In section 3.1 they do not indicate whether there is legislation that regulates the location of landfills with respect to the distance to: surface water bodies, groundwater, population centers, airports, outside flood zones, fault zones, unstable slopes, etc.

12.- Section 3.2 should include generation, that is, have the title: “generation and composition of waste” and mention the per capita and total generation estimated in the study area.

13.- In section 3.2 the percentages that indicate are not clear: “…Almost all the landfills found consisted of household waste (97%), garden waste (91%) and construction waste (66%) ...”

14.- In section 3.3, it is not necessary to indicate for how many years the problem of illegal dumping has been present in the study area.

15. In section 3.5 there is repeated information in different lines. For example, in line 249 and line 254 “…Lack of awareness…”

16.- It is suggested that the wording of paragraph 3.5 be carried out in accordance with the order of table 3, because it is confusing (Correct from the third factor reported onwards).

17.- Paragraph 3.6 must address each and every one of the impacts shown in table 4.

18.- In section 3.6, “…smells…” is repeated twice. Check the wording.

19.- With the paragraph "...Most of the respondents (70%) felt that ISWD reduced the economic value..." confirms that there are also economic impacts, which reinforces observation 2.

20.- It is necessary to cite the table in paragraph 3.6

4. Discussion

21.- Why are only two topics addressed in this discussion section, when seven were presented in the results?

     3.1. Abundance, distribution and size of illegal solid waste dumps

     3.2. Composition of solid waste

     3.3. The socio-demographic profile of the respondents

     3.4. Solid waste disposal practices

     3.5. Factors contributing to illegal solid waste dumping

     3.6. Perceived impacts of illegal solid waste dumping

     3.7. Potential interventions reported by respondents.

4.1. Distribution and composition of illegal solid waste dumps.

23.- The paragraphs of line 311 and 316 are repeated, so it is necessary to review the wording to join them.

24.- On line 326 they indicate “…hazardous waste to health…” but they do not specify which can be found in municipal solid waste.

4.3. Factors that influence the illegal dumping of solid waste

25.- Somewhere in the manuscript it should be mentioned what is provided for in local legislation, regarding the responsibility of the authorities to provide the waste management service.

26. Lines 371, 372 and 373 indicate that the municipal failure can be attributed to incompetence and rampant corruption, but surely financial resources also constitute another problem.

27. In this section, informal groups of collectors are not mentioned, which, as their name indicates, are those people who participate in the recovery of by-products to obtain income and their participation is important for management proposals.

28.- Lines 422 and 423 indicate "...raising awareness and educating people..." so it should be reinforced by mentioning dissemination, dissemination and training programs.

29.- In lines 430 and 431 they mention that "...most waste generated by inhabitants of developing countries is recyclable..." and it is not clear if they are including, in addition to paper, cardboard, glass, plastics, metals etc to organic matter, so it is important to clarify it in section 3.2, regarding the percentages that make up waste organic matter, recyclable and difficult-to-recycle materials or others.

5. Conclusions

30.- Somewhere it is necessary to mention that the municipality must have a municipal program for the prevention and comprehensive management of waste, which includes all the required strategies.

31. It remains to mention the importance of incorporating informal groups of collectors.

Author Response

Please, see attached file.

Reviewer 5 Report

The paper is written clearly, the state of art, methodology and results are presented logical and understandable for readers and but conclusion is so laconic, what is a recommendation for solving the problem?

Please avoid dumping citation for example line 93, 307.

Author Response

Please, see attached file.

Round 2

Reviewer 2 Report

The author revised and improved the article carefully according to the requirements of the reviewers, and did a lot of work. The readability of the article has been significantly improved. I am satisfied with the revised version and the author’s reply. I think it can be accepted and published.